# Effects of gene–lifestyle interactions on obesity based on a multi-locus risk score: A cross-sectional analysis

Sho Nakamura[1,2]*, Xuemin Fang[1], Yoshinobu Saito[2,3,4], Hiroto Narimatsu[1,2,3,5], Azusa Ota[6], Hiroaki Ikezaki[6,7], Chisato Shimanoe[8], Keitaro Tanaka[9], Yoko Kubo[10], Mineko Tsukamoto[10], Takashi Tamura[10], Asahi Hishida[10], Isao Oze[11], Yuriko N. Koyanagi[12], Yohko Nakamura[13], Miho Kusakabe[13], Toshiro Takezaki[14], Daisaku Nishimoto[15], Sadao Suzuki[16], Takahiro Otani[16], Nagato Kuriyama[17,18], Daisuke Matsui[17], Kiyonori Kuriki[19], Aya Kadota[20], Yasuyuki Nakamura[20,21], Kokichi Arisawa[22], Sakurako Katsuura-Kamano[22], Masahiro Nakatochi[23], Yukihide Momozawa[24], Michiaki Kubo[24], Kenji Takeuchi[10,25], Kenji Wakai[10]

1 Graduate School of Health Innovation, Kanagawa University of Human Services, Kawasaki, Kanagawa, Japan, 2 Cancer Prevention and Control Division, Kanagawa Cancer Center Research Institute, Yokohama, Kanagawa, Japan, 3 Center for Innovation Policy, Kanagawa University of Human Services, Kawasaki, Kanagawa, Japan, 4 Faculty of Sport Management, Nippon Sport Science University, Yokohama, Kanagawa, Japan, 5 Department of Genetic Medicine, Kanagawa Cancer Center, Yokohama, Kanagawa, Japan, 6 Department of General Internal Medicine, Kyushu University Hospital, Fukuoka, Fukuoka, Japan, 7 Department of Comprehensive General Internal Medicine, Faculty of Medical Sciences, Fukuoka, Fukuoka, Japan, 8 Department of Pharmacy, Saga University Hospital, Nabeshima, Saga, Japan, 9 Department of Preventive Medicine, Faculty of Medicine, Saga University, Nabeshima, Saga, Japan, 10 Department of Preventive Medicine, Nagoya University Graduate School of Medicine, Nagoya, Aichi, Japan, 11 Division of Cancer Epidemiology and Prevention, Aichi Cancer Center Research Institute, Nagoya, Aichi, Japan, 12 Division of Cancer Information and Control, Aichi Cancer Center Research Institute, Nagoya, Aichi, Japan, 13 Cancer Prevention Center, Chiba Cancer Center Research Institute, Chiba, Chiba, Japan, 14 Department of International Island and Community Medicine, Kagoshima University Graduate School of Medical and Dental Sciences, Sakuragaoka, Kagoshima, Japan, 15 School of Health Sciences, Faculty of Medicine, Kagoshima University, Kagoshima, Kagoshima, Japan, 16 Department of Public Health, Nagoya City University Graduate School of Medical Sciences, Nagoya, Aichi, Japan, 17 Department of Epidemiology for Community Health and Medicine, Kyoto Prefectural University of Medicine, Nagoya, Aichi, Japan, 18 Shizuoka Graduate University of Public Health, Shizuoka, Shizuoka, Japan, 19 Laboratory of Public Health, Division of Nutritional Sciences, School of Food and Nutritional Sciences, University of Shizuoka, Shizuoka, Shizuoka, Japan, 20 NCD Epidemiology Research Center, Shiga University of Medical Science, Otsu, Shiga, Japan, 21 Takeda Hospital Medical Examination Center, Kyoto, Kyoto, Japan, 22 Department of Preventive Medicine, Tokushima University Graduate School of Biomedical Sciences, Tokushima, Tokushima, Japan, 23 Public Health Informatics Unit, Department of Integrated Health Sciences, Nagoya University Graduate School of Medicine, Nagoya, Aichi, Japan, 24 Laboratory for Genotyping Development, RIKEN Center for Integrative Medical Sciences, Yokohama, Kanagawa, Japan, 25 Department of International and Community Oral Health, Tohoku University Graduate School of Dentistry, Sendai, Miyagi, Japan

* research@nakasho.org



**Data Availability Statement:** Data cannot be shared publicly because of ethical reasons. Data described in the manuscript will be made available upon application and approval from the J-MICC

## Abstract

### Background

The relationship between lifestyle and obesity is a major focus of research. Personalized nutrition, which utilizes evidence from nutrigenomics, such as gene–environment interactions, has been attracting attention in recent years. However, evidence for gene–

study group (http://cohort.umin.jp/english/form/index.html) for researchers who meet the criteria for access to confidential data.

**Funding:** This study was supported by Grants-in-Aid for Scientific Research for Priority Areas of Cancer (No. 17015018) and Innovative Areas (No. 221S0001) and by the Japan Society for the Promotion of Science (JSPS) KAKENHI Grant (No. 16H06277 [CoBiA]) from the Japanese Ministry of Education, Culture, Sports, Science and Technology. This work was also supported in part by funding for the BioBank Japan Project from the Japan Agency for Medical Research and Development since April 2015, and the Ministry of Education, Culture, Sports, Science and Technology from April 2003 to March 2015. There was no additional external funding received for this study. The funders had no role in study design, data collection and analysis, decision to publish, or preparation of the manuscript.

**Competing interests:** The authors have declared that no competing interests exist.

environment interactions that can inform treatment strategies is lacking, despite some reported interactions involving dietary intake or physical activity. Utilizing gene–lifestyle interactions in practice could aid in optimizing interventions according to genetic risk.

## Methods

This study aimed to elucidate the effects of gene–lifestyle interactions on body mass index (BMI). Cross-sectional data from the Japan Multi-Institutional Collaborative Cohort Study were used. Interactions between a multi-locus genetic risk score (GRS), calculated from 76 ancestry-specific single nucleotide polymorphisms, and nutritional intake or physical activity were assessed using a linear mixed-effect model.

## Results

The mean (standard deviation) BMI and GRS for all participants (n = 12,918) were 22.9 (3.0) kg/m$^2$ and -0.07 (0.16), respectively. The correlation between GRS and BMI was $r$ (12,916) = 0.13 (95% confidence interval [CI] 0.11–0.15, $P$ < 0.001). An interaction between GRS and saturated fatty acid intake was observed (β = -0.11, 95% CI -0.21 to -0.02). An interaction between GRS and $n$-3 polyunsaturated fatty acids was also observed in the females with normal-weight subgroup (β = -0.12, 95% CI -0.22 to -0.03).

## Conclusion

Our results provide evidence of an interaction effect between GRS and nutritional intake and physical activity. This gene–lifestyle interaction provides a basis for developing prevention or treatment interventions for obesity according to individual genetic predisposition.

## Introduction

Obesity is one of the leading causes of death by increasing the risk of non-communicable diseases, such as cardiovascular diseases, type-2 diabetes, musculoskeletal disorders, and cancers [1]. The prevalence of obesity is increasing worldwide, necessitating the development of more effective prevention and treatment strategies. Both environmental and genetic factors are responsible for obesity. Environmental factors include diet, exercise, and the obesogenic environment, including socioeconomic, ethnic, cultural, and geographical factors; notably, some of these factors are beyond an individual's control [2, 3]. In addition, an obesogenic environment could affect lifestyle factors, such as diet and exercise.

The independent effects of these factors on body mass index (BMI) are limited. Rather, combinations and interactions between environmental and genetic factors have a relatively large effect on BMI. Gene–environment interactions contributing to BMI have therefore attracted substantial attention [4–6]; the effect of environmental factors on BMI is modified by genetic factors. Recent research aims to identify gene–environment interactions that can serve as a basis for the prevention or treatment of obesity. However, the results of these studies have been contradictory and inconsistent, despite some evidence indicating that the effects of dietary intake, physical activity, or socioeconomic status on BMI differ according to genetic risk [7–11]. The existence of gene–environment interactions indicates the importance of utilizing gene–lifestyle interactions in primary preventive intervention to select optimal intervention

according to genetic risk. For example, personalized nutrition based on nutrigenomic evidence gives specific dietary advice to each individual based on one's genetic information, rather than a general recommendation about diet [12].

In this study, we focused on the genetic risk score (GRS) calculated from multiple single nucleotide polymorphisms (SNPs) or loci (multi-locus GRS), diet, and physical activity to analyze gene–lifestyle interactions. Data are limited regarding the marginal effects of interactions when including nutrients, such as carbohydrates, fat, protein, dietary fiber, vitamins, and physical activity in a single statistical model. Such a gene–lifestyle interaction would help advance the utilization of genetic risk in managing obesity since diet and physical activity are the two major components of lifestyle interventions for obesity. This study aimed to explore the interaction between genetic risk and lifestyle factors, such as nutritional intake and physical activity, on BMI using cross-sectional data from the Japan Multi-Institutional Collaborative Cohort Study (J-MICC Study).

## Materials and methods

### Study population

Data were obtained from the J-MICC Study (ver. 20190720), a genomic cohort study launched in 2005. Details of the J-MICC study are described elsewhere [13, 14]. In brief, participants were residents of Japan who participated in health checkups held by local governments, volunteers, or patients recruited at their first visit to the cancer hospital, aged 35 to 69 years. Information on lifestyle and medical conditions were collected by a self-administered questionnaire, and various parameters, such as anthropometric traits and laboratory data from blood samples, including genomic information, were obtained in the baseline survey. Participants were recruited from 12 different areas throughout Japan (Aichi, Chiba, Fukuoka, Kagoshima, Kyoto, Kyushu-KOPS, Okazaki, Sakuragaoka, Saga, Shizuoka-Daiko, Takashima, and Tokushima) between 2004 and 2013. Written informed consent was obtained from all participants. The J-MICC Study was approved by the Ethics Committees of Nagoya University Graduate School of Medicine and the other institutions participating in the J-MICC study. This analysis was also approved by the ethics committee of the Kanagawa Cancer Center.

### Genotyping, quality control, and genotype imputation

DNA was extracted from the buffy coat using a BioRobot M48 Workstation (QIAGEN Group, Tokyo, Japan). SNP genotyping was performed by the RIKEN Center for Integrative Medical Sciences using the Illumina OmniExpressExome Array (Illumina, San Diego, CA, USA). Inconsistencies in sex information between the questionnaire and genotype-based estimates were excluded (n = 26). The identity-by-descent method implemented in PLINK 1.9 found 388 closely related pairs (pi-hat > 0.1875), and one sample in each pair was excluded [15, 16]. Subjects whose ancestries were estimated to be outside of the Japanese population detected by a principal component analysis with a 1000 Genomes reference panel (phase 3) were excluded (n = 34) [17–19]. SNPs with a genotype call rate of <0.98 and/or a Hardy–Weinberg equilibrium exact test p-value of $<1 \times 10^{-6}$, a low minor allele frequency (MAF) of <0.01, or an allele frequency difference > 20% between the scaffold and 1000 genomes phase 3 EAS (East Asian) samples were excluded. Quality control filtering resulted in 14,086 individuals and 570,162 SNPs [20]. Genotype imputation was performed using SHAPEIT version 2 and Minimac3 based on the 1000 Genomes Project (phase 3) as a cosmopolitan reference panel [21, 22].

## Genetic risk score

SNPs used to calculate the GRS were selected from a genome-wide association study (GWAS) of the Japanese population reported by Akiyama et al. [23]. Details including the MAF and $r^2$ of the selected 76 SNPs are shown in S1 Table. SNPs on chromosome X and SNPs identified by a sex-stratified analysis were excluded. The GRS was calculated from the β coefficients of these 76 SNPs using a previously reported weighting method [7, 8, 23]. In brief, GRS was calculated by summing the allelic dosage (0 to 2) for each SNP, which was weighted by the β coefficient reported in GWAS [2]. GRS was constructed with a theoretical range of -2.256 (possessing no predisposition allele) to 2.334 (possessing 152 predisposition alleles), where higher scores indicate a higher genetic predisposition to obesity. The calculated GRS was categorized into lower and upper groups according to the score in order to make the number of participants approximately equal in each group.

## Phenotypes

Weight and height were measured at baseline and were used to calculate BMI (kg/m$^2$). Missing values were complemented by BMI calculated using the self-reported height and weight (questionnaire). Daily nutritional intake and physical activity were assessed using the J-MICC Study questionnaire at baseline. The questionnaires were reviewed by trained staff for their credibility and consistency. Daily nutritional intake was assessed using a validated food frequency questionnaire (S1 Fig) [24–26]. Nutritional intake was adjusted by total energy intake using the residual method. Daily physical activity was calculated, based on a previously reported method, in terms of metabolic equivalents-hours per day (METs-h/day) [27]. In brief, daily life activity and leisure-time activity were estimated based on the International Physical Activity Questionnaire [28]. The intensity of activity was categorized into five levels: walking, 3.0; and heavy physical work or exercise, 4.5 METs in daily life activities and 3.4, 7.0, and 10.0 METs for leisure-time activities. The intensity was multiplied by the length and frequency of each activity to obtain METs-h/day. We did not include standing time (2.0 METs) as we intended to assess moderate-to-vigorous physical activity (> 3 METs) in our analysis. Sitting time (h/day) was obtained from the questionnaire. Information for all parameters was available for 13,913 participants.

## Statistical analysis

Statistical analyses were performed using R version 3.6.3 [29]. Pearson's product-moment correlation between GRS and BMI was calculated, and a simple linear regression analysis was performed. A linear mixed-effects model (LMM) was constructed and fit by maximum likelihood using lme4 version 1.1-23 [30]. The effective degrees of freedom were approximated using the Welch-Satterthwaite method implemented in lmerTest version 3.1-2 [31]. The dependent variable was BMI with a recruited site-specific random intercept, and the fixed effects were GRS (lower and upper halves were coded 0 and 1, respectively), age, sex, BMI measurement method (estimates based on examined or self-reported were coded 0 and 1, respectively), 21 lifestyle factors, interaction terms between age and sex and between GRS and age, sex, and lifestyle factors. The interaction terms were assessed for their effects on BMI. The 21 lifestyle factors included daily nutritional intake (energy, protein, saturated fatty acids, monounsaturated fatty acids, $n$-3 polyunsaturated fatty acids, $n$-6 polyunsaturated fatty acids, carbohydrates, soluble dietary fiber, insoluble dietary fiber, retinol, vitamin D, vitamin E, vitamin B1, vitamin B2, folate, vitamin C, iron, and calcium), alcohol intake, MVPA, and sitting time. Continuous variables other than BMI and GRS were standardized based on z-score. Additionally, 995/13,913 (7.2%) participants were excluded owing to influential data points according to the residual, leverage, and Cook's distance. Variable selection was performed by backward elimination with

an alpha level of 0.05, partly using the step function in lmerTest [31]. Visual inspection of residual plots did not reveal any obvious deviations from homoscedasticity or normality. We performed a sensitivity analysis excluding the 1,082 participants recruited at the Cancer Center (Aichi 1, S2 Table). A subgroup analysis was also performed according to BMI (normal weight [>18.5 kg/m$^2$, <25 kg/m$^2$] or obesity [≥25 kg/m$^2$]) and sex; subgroups 1–4 corresponded to males with normal weight, females with normal weight, males with obesity, and females with obesity, respectively. As sex was used as stratifying factor, it was dismissed in the subgroup analysis. Monounsaturated fatty acids and vitamin E in subgroup 2 were excluded from the model owing to multicollinearity. Variance inflation factors (VIF) were checked for multicollinearity using the vif function in car version 3.0-3 [32], and the highest VIF among all variables in 5 models was 3.4, indicating no substantial influence of multicollinearity on model results. The gene–lifestyle interaction was further explored using a different approach: candidate lifestyle factors were explored in the subgroup analysis according to GRS, followed by interaction analysis factor by factor (candidate approach). The detail of the candidate approach is described in detail in the S1 Appendix.

## Results

### Characteristics of the participants

Demographic, lifestyle, and genetic factors for study participants are shown in Table 1, including characteristics according to GRS subgroups used to generate covariance matrixes in the LMM. Participants were recruited from 12 sites, ranging from 466/12,918 participants (3.6%) at the smallest site to 1,888/12,918 participants (14.6%) (S2 Table). The mean (standard deviation) BMI for all participants was 22.9 (3.0) kg/m$^2$, for male was 23.6 (2.8) kg/m$^2$, and for female was 22.2 (3.0) kg/m$^2$. In subgroups, there were 9,173 participants of normal weight (male/female 3,952/5,221) and 2,945 participants with obesity (male/female, 1,683/1,262) (S3 Table). There were no statistical differences in age, sex distribution, BMI, or the captured environmental variables between the two subgroups of the GRS (Table 1). The correlation between GRS and BMI was $rho$(12,916) = 0.13 (95% confidence interval [CI] 0.11–0.15, $p < 0.001$), and the increase in BMI for every unit increase in the GRS was 2.45 (95% CI 2.13–2.77, $p < 0.001$).

### Gene–lifestyle interaction analysis

The results of the gene–lifestyle interaction analysis are summarized in Table 2. The final LMM included 15 predictors of relevance selected via stepwise regression with backward selection: GRS, age, sex, 8 nutritional factors, sitting time, and interaction terms for age × sex, GRS × age, and GRS × saturated fatty acids as fixed effects. BMI was approximated using the following equation:

$$Y_{ij} = 23.3 + \sum_{j=1}^{11}(\theta_j + \gamma_j \times site_j) + (0.63 \times \mathrm{GRS}_{ij}) + \sum_{k=1}^{11}(\beta_k \times X_{kij}) + (0.56 \times [\mathrm{age}_{ij} * \mathrm{sex}_{ij}])$$
$$+ (-0.10 \times [\mathrm{GRS}_{ij} * \mathrm{age}_{ij}]) + (-0.11 \times [\mathrm{GRS}_{ij} * \text{saturated fatty acids}_{ij}]) + \epsilon_{ij,}$$

where $Y_{ij}$ = BMI (kg/m$^2$), $\theta_j$ is the random intercept for the j$^{th}$ site dummy variable, $\gamma_j$ is the coefficient for the j$^{th}$ site dummy variable, $\beta_k$ = coefficient for fixed effect $k$, $X_{kij}$ = value of the fixed effect $k$ for participant $I$ at site $j$, $\epsilon_{ij}$ = residual for participant $I$ in recruited site $j$; and fixed effect $k$ is either age, sex, protein, saturated fatty acids, $n$-3 polyunsaturated fatty acids, carbohydrate, soluble dietary fiber, retinol, vitamin D, vitamin B1 or sitting time. There were differences in BMI between the recruited sites with a variance of 0.39 kg/m$^2$. The only interaction term with GRS and lifestyle factor that remained after the variable selection procedure was saturated fatty acids intake (Table 2). The interaction between GRS and saturated fatty

**Table 1. Characteristics of the study participants.**

| Characteristics | | Subgroups, according to GRS | |
| --- | --- | --- | --- |
| | **All participants** | **Lower half** | **Upper half** |
| | n = 12,918 | n = 6,461 | n = 6,457 |
| **Age (years)** | 54.7 ± 9.3 | 54.8 ± 9.3 | 54.6 ± 9.3 |
| **Sex (F, %)** | 7126 ± 55.2 | 3545 ± 54.9 | 3581 ± 55.5 |
| **BMI (kg/m$^2$)** | 22.9 ± 3.0 | 22.6 ± 2.9 | 23.2 ± 3.1 |
| **Measurement method (self-report, %)** | 2549 ± 19.7 | 1279 ± 19.8 | 1270 ± 19.7 |
| **Daily nutritional intakes** | | | |
| **Energy (kcal)** | 1685.1 ± 336.2 | 1686.6 ± 335.4 | 1683.6 ± 337.1 |
| **Protein (g)** | 52.7 ± 7.0 | 52.6 ± 6.9 | 52.8 ± 7.1 |
| **Saturated fatty acids (g)** | 11.2 ± 2.5 | 11.1 ± 2.5 | 11.2 ± 2.4 |
| **Monounsaturated fatty acids (g)** | 16.0 ± 3.5 | 16.0 ± 3.4 | 16.1 ± 3.6 |
| ***n*-3 polyunsaturated fatty acids (g)** | 2.2 ± 0.5 | 2.2 ± 0.5 | 2.2 ± 0.5 |
| ***n*-6 polyunsaturated fatty acids (g)** | 10.9 ± 2.7 | 10.8 ± 2.7 | 10.9 ± 2.8 |
| **Carbohydrate (g)** | 240.6 ± 23.4 | 241.0 ± 23.4 | 240.2 ± 23.3 |
| **Total dietary fiber (g)** | 10.5 ± 2.8 | 10.5 ± 2.8 | 10.5 ± 2.8 |
| **Soluble dietary fiber (g)** | 1.9 ± 0.6 | 1.9 ± 0.6 | 1.9 ± 0.6 |
| **Insoluble dietary fiber (g)** | 7.6 ± 2.0 | 7.6 ± 2.0 | 7.6 ± 2.0 |
| **Retinol (mcg)** | 923.7 ± 360.6 | 919.7 ± 360.4 | 927.6 ± 360.9 |
| **Vitamin D (mcg)** | 7.1 ± 2.9 | 7.1 ± 2.9 | 7.1 ± 2.9 |
| **Vitamin E (mg)** | 8.0 ± 1.8 | 8.0 ± 1.8 | 8.0 ± 1.8 |
| **Vitamin B1 (mg)** | 0.6 ± 0.1 | 0.6 ± 0.1 | 0.6 ± 0.1 |
| **Vitamin B2 (mg)** | 1.1 ± 0.2 | 1.1 ± 0.2 | 1.1 ± 0.2 |
| **Folate (mcg)** | 326.0 ± 94.6 | 325.2 ± 93.7 | 326.7 ± 95.6 |
| **Vitamin C (mg)** | 94.3 ± 33.8 | 94.2 ± 33.5 | 94.5 ± 34.0 |
| **Calcium (mg)** | 505.1 ± 139.2 | 504.0 ± 140.2 | 506.3 ± 138.1 |
| **Iron (mg)** | 6.9 ± 1.7 | 6.9 ± 1.7 | 6.9 ± 1.7 |
| **Ethanol (g)** | 13.4 ± 22.5 | 13.2 ± 22.2 | 13.5 ± 22.8 |
| **Physical activity (METs-h/day)** | 13.7 ± 12.5 | 13.7 ± 12.4 | 13.6 ± 12.6 |
| **Sitting time (hours/day)** | 4.9 ± 3.7 | 4.9 ± 3.7 | 4.9 ± 3.7 |
| **GRS** | -0.07 ± 0.16 | -0.20 ± 0.09 | 0.06 ± 0.10 |

Data are presented as means ± standard deviation, unless otherwise specified. GRS, genetic risk score; g, grams; mcg, micrograms; METs-h/day, metabolic equivalents-hours per day.

acid intake ($P$ = 0.021) is described in Fig 1. A negative association between saturated fatty acid intake and GRS was exaggerated in participants with a high GRS subgroup (upper half). The differences in BMI associated with 10 grams (4.0 standardized units) of saturated fatty acids per day were 0.27 kg/m$^2$ and 0.74 kg/m$^2$ in the low and high GRS subgroups, respectively. A comparable result was obtained from the sensitivity analysis (S4 Table).

## Subgroup analysis according to BMI and sex

The results of a subgroup analysis are shown in Table 3, and the results of the interaction analysis are graphically presented in Fig 2. Among the lifestyle factors, gene–lifestyle interactions were observed for n-3 polyunsaturated fatty acids, vitamin B1, and sitting time. These interactions were prevalent only in subgroups of female participants. Notably, the association between lifestyle factors and GRS was exaggerated in the high GRS subgroups compared with the low GRS subgroup. Association between sitting time and BMI was only observed for the

**Table 2. Effects of gene–lifestyle interactions on BMI.**

| Parameters | | | |
|---|---|---|---|
| **Random effects** | **Variance** | **Standard deviation** | |
| **Recruited sites (intercept)** | 0.39 | 0.62 | |
| **Residual** | 7.81 | 2.78 | |
| **Fixed effects** | **Coefficient estimate** | **95% confidence interval** | ***p*-value** |
| **Intercept** | 23.34 | 22.98, 23.70 | < 0.001 |
| **GRS (high GRS subgroup)** | 0.63 | 0.54, 0.73 | < 0.001 |
| **Age** | 0.02 | -0.08, 0.11 | 0.738 |
| **Sex (female)** | -1.35 | -1.46, -1.24 | < 0.001 |
| **Protein** | 0.20 | 0.10, 0.30 | < 0.001 |
| **Saturated fatty acids** | -0.07 | -0.16, 0.02 | 0.151 |
| ***n*-3 polyunsaturated fatty acids** | 0.21 | 0.14, 0.29 | < 0.001 |
| **Carbohydrate** | 0.08 | 0.02, 0.14 | 0.008 |
| **Soluble dietary fiber** | -0.29 | -0.35, -0.22 | < 0.001 |
| **Retinol** | 0.09 | 0.03, 0.14 | 0.003 |
| **Vitamin D** | -0.19 | -0.27, -0.10 | < 0.001 |
| **Vitamin B1** | 0.07 | 0.01, 0.13 | 0.033 |
| **Sitting time** | 0.08 | 0.02, 0.15 | 0.012 |
| **Age * sex** | 0.56 | 0.46, 0.66 | < 0.001 |
| **Genetic risk score * age** | -0.10 | -0.20, -0.002 | 0.045 |
| **Genetic risk score * saturated fatty acids** | -0.11 | -0.21, -0.02 | 0.021 |

Continuous variables other than BMI and GRS were standardized based on z-score. Variables selected by the backward reduction from the following fixed effects are shown: GRS (low and high GRS subgroups coded as 0 and 1), age, sex, BMI measurement method (calculated from examined or self-reported height and weight coded as 0 and 1), energy, protein, saturated fatty acids, monounsaturated fatty acids, n-3 polyunsaturated fatty acids, n-6 polyunsaturated fatty acids, carbohydrate, soluble dietary fiber, insoluble dietary fiber, retinol, vitamin D, vitamin E, vitamin B1, vitamin B2, folate, vitamin C, iron, and calcium, alcohol intake, moderate-to-vigorous physical activity, sitting time, interaction terms between age and sex, and GRS and age, sex, and each lifestyle factor. GRS, genetic risk score.

high GRS subgroup in females with obesity (subgroup 2). Variances of BMI between the recruited sites were largest in this subgroup. No gene-lifestyle interaction was observed for subgroup 1, while the association between MVPA and BMI was only observed for this subgroup.

## Results of the candidate approach

The LMM for the subgroups in the first to fourth quartiles of the GRS (details in S1 Appendix) is shown in S5 Table. Candidate variables for the interaction analysis were protein, saturated fatty acids, *n*-3 polyunsaturated fatty acids, *n*-6 polyunsaturated fatty acids, carbohydrates, retinols, vitamins D, E, B1, and calcium. The β-coefficients and *P*-values for the interaction term between GRS and each candidate variable are shown in S6 Table and S2 Fig. Calcium intake showed the strongest evidence for the existence of gene–lifestyle interaction (*P* = 0.0031), followed by saturated fatty acids (*P* = 0.0078).

## Discussion

In this study, the gene–lifestyle interaction between GRS, calculated from 76 SNPs known to be related to BMI based on an ancestry-specific GWAS, and broad lifestyle factors, such as nutrition intake and physical activity, were assessed simultaneously in a single model. The

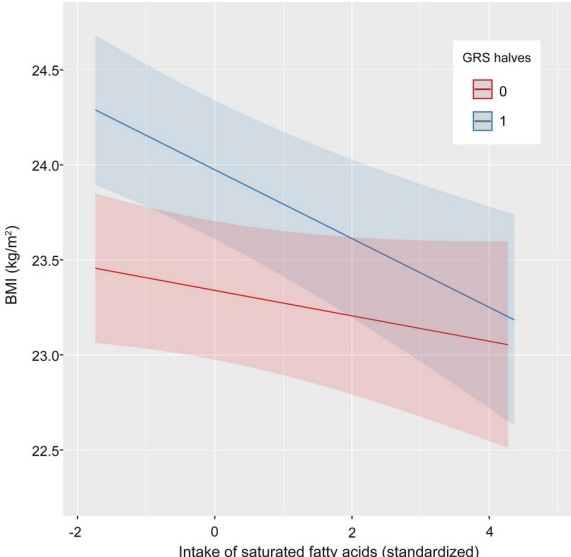

**Fig 1. Graphical representation of the interaction between GRS and saturated fatty acid intake.** Red and blue lines indicate regression lines for individuals with GRS values in the lower half (GRS = 0) and upper half (GRS = 1). The shading around each regression line shows the 95% confidence interval. One unit of standardized saturated fatty acids intake (*x*-axis) corresponds to 2.5 g/day. The association between BMI and saturated fatty acid intake is greater in the group with a high GRS. GRS, genetic risk score.

existence of gene–lifestyle interactions in obesity has been controversial. However, our results provide evidence for an effect of a gene–lifestyle interaction on the BMI phenotype.

Interactions between GRS and saturated fatty acids were observed in our analysis of all participants, including lean and obese participants. The mean intake of saturated fatty acids was 11.1, and 11.2 grams per day in high, and low GRS groups respectively, whereas for participants reporting low saturated fatty acid intake, BMI was higher in the high GRS group. This may be interpreted as reverse causation based on research in nutrition science, rather than an effect of saturated fatty acid intake on BMI. In other words, individuals who possess a higher BMI tend to abstain from saturated fatty acid intake (e.g., milk, beef, and pork [S1 Fig]). Results contradicting the results from prior studies [33, 34] might be due to the differences in ethnicity. Another reason might be the nature of the partial regression coefficient obtained from the analysis, while other predictors were held constant. Although this is the same for other predictor variables, individuals who possess a high GRS and higher BMI could focus on factors associated with BMI described in Table 2 other than saturated fatty acids, such as decreasing energy intake (protein, fats, and carbohydrate), increasing soluble dietary fiber intake and decreasing sitting time, because these are consistent with the known evidence [35, 36]. Incorporating a dietary pattern, which is also indicated in the prior literature, in the future study is an idea to solve challenges in interpreting the partial regression coefficient.

This study had a few key strengths. For example, whereas most studies of gene–lifestyle interactions have focused on specific genes [4, 5], which may be sufficient for single-gene disorders, we used a GRS involving multiple loci, which is more appropriate for multifactorial diseases like obesity. Accounting for many SNPs can lead to a more precise evaluation of the disease risk [23, 37]. In addition, we calculated GRS based on a GWAS of a population with the same ancestry as the study population [23]. GWAS results often differ depending on populations [38]; however, these differences have not been considered in some previous studies of gene–lifestyle interactions [4]. We observed a gene–lifestyle interaction in subgroup 4 (females

**Table 3. Subgroup analysis of effects of gene–lifestyle interactions on BMI.**

| Parameters | Subgroup 1 (Normal weight, male) | | | Subgroup 2 (Normal weight, female) | | | Subgroup 3 (Obese, male) | | | Subgroup 4 (Obese, female) | | |
|---|---|---|---|---|---|---|---|---|---|---|---|---|
| Number of participants | n = 3,952 | | | n = 5,221 | | | n = 1,683 | | | n = 1,262 | | |
| Random effects | Variance | Standard deviation | | Variance | Standard deviation | | Variance | Standard deviation | | Variance | Standard deviation | |
| Recruited sites (intercept) | 0.03 | 0.18 | | 0.10 | 0.32 | | 0.03 | 0.17 | | 0.04 | 0.21 | |
| Residual | 2.50 | 1.58 | | 2.65 | 1.63 | | 2.75 | 1.66 | | 2.74 | 1.66 | |
| Fixed effects | Estimate | 95% CI | p-value | Estimate | 95% CI | p-value | Estimate | 95% CI | p-value | Estimate | 95% CI | p-value |
| Intercept | 22.35 | 22.22, 22.47 | < 0.001 | 21.56 | 21.38, 21.75 | < 0.001 | 26.83 | 26.67, 26.98 | < 0.001 | 26.90 | 26.71, 27.09 | < 0.001 |
| GRS (high GRS subgroup) | 0.21 | 0.11, 0.31 | < 0.001 | 0.21 | 0.12, 0.30 | < 0.001 | 0.32 | 0.16, 0.48 | < 0.001 | 0.22 | 0.03, 0.40 | 0.022 |
| Age (years) | 0.10 | 0.05, 0.15 | < 0.001 | 0.27 | 0.22, 0.32 | < 0.001 | 0.04 | -0.09, 0.16 | 0.652 | - | | |
| Protein | - | | | - | | | - | | | 0.14 | 0.02, 0.25 | 0.018 |
| Saturated fatty acids | 0.07 | 0.01, 0.12 | 0.01 | - | | | - | | | - | | |
| n-3 polyunsaturated fatty acids | - | | | 0.03 | -0.06, 0.11 | 0.524 | - | | | - | | |
| n-6 polyunsaturated fatty acids | - | | | 0.12 | 0.05, 0.19 | < 0.001 | - | | | - | | |
| Carbohydrate | - | | | 0.07 | 0.02, 0.12 | 0.003 | - | | | 0.14 | 0.04, 0.24 | 0.008 |
| Soluble dietary fiber | -0.10 | -0.16, -0.03 | -0.16 | - | | | -0.13 | -0.22, -0.04 | 0.005 | -0.14 | -0.24, -0.03 | 0.011 |
| Insoluble dietary fiber | - | | | -0.12 | -0.17, -0.07 | < 0.001 | - | | | - | | |
| Retinol | 0.06 | 0.002, 0.11 | 0.002 | - | | | - | | | - | | |
| Vitamin E | 0.10 | 0.04, 0.16 | 0.04 | - | | | - | | | - | | |
| Vitamin B1 | - | | | 0.04 | -0.03, 0.11 | 0.308 | - | | | - | | |
| Calcium | - | | | - | | | -0.12 | -0.21, -0.03 | 0.012 | - | | |
| Moderate-to-vigorous physical activity | -0.07 | -0.12, -0.01 | -0.12 | - | | | - | | | - | | |
| Sitting time | - | | | - | | | - | | | -0.03 | -0.18, 0.13 | 0.757 |
| GRS * age | - | | | - | | | -0.26 | -0.43, -0.10 | 0.001 | - | | |
| GRS * n-3 polyunsaturated fatty acids | - | | | -0.12 | -0.22, -0.03 | 0.011 | - | | | - | | |
| GRS * vitamin B1 | - | | | 0.10 | 0.01, 0.20 | 0.035 | - | | | - | | |
| GRS * sitting time | - | | | - | | | - | | | 0.20 | 0.01, 0.39 | 0.036 |

Continuous variables other than BMI and GRS were standardized based on z-score. Variables selected by the backward reduction from the following fixed effects are shown: GRS (lower and upper halves coded as 0 and 1), age, BMI measurement method (calculated from examined or self-reported height and weight coded as 0 and 1), energy, protein, saturated fatty acids, monounsaturated fatty acids, n-3 polyunsaturated fatty acids, n-6 polyunsaturated fatty acids, carbohydrate, soluble dietary fiber, insoluble dietary fiber, retinol, vitamin D, vitamin E, vitamin B1, vitamin B2, folate, vitamin C, iron, and calcium, alcohol intake, moderate-to-vigorous physical activity, sitting time, interaction terms between GRS and age, and each lifestyle factor. Hyphens indicate variables that were eliminated in the variable selection procedure. GRS, genetic risk score; CI, confidence interval.

[a]Subgroup 1; male, BMI >18.5 $kg/m^2$”nd <‘5 $kg/m^2$: Subgroup 2; female BMI >18.5 $kg/m^2$ and <25 $kg/m^2$: Subgroup 3; male BMI ≥25 $kg/m^2$: Subgroup 4; female BMI ≥25 $kg/m^2$.

with obesity), while the sample size in this subgroup was the smallest among all four subgroups. This suggests that the use of SNPs and beta coefficients reported in the GWAS on the Japanese population contributed to observing the gene–environment interaction in the smallest subgroup. Moreover, the results of the subgroup analyses are important for indicating an interaction effect for the groups that are classified with a combination of population group

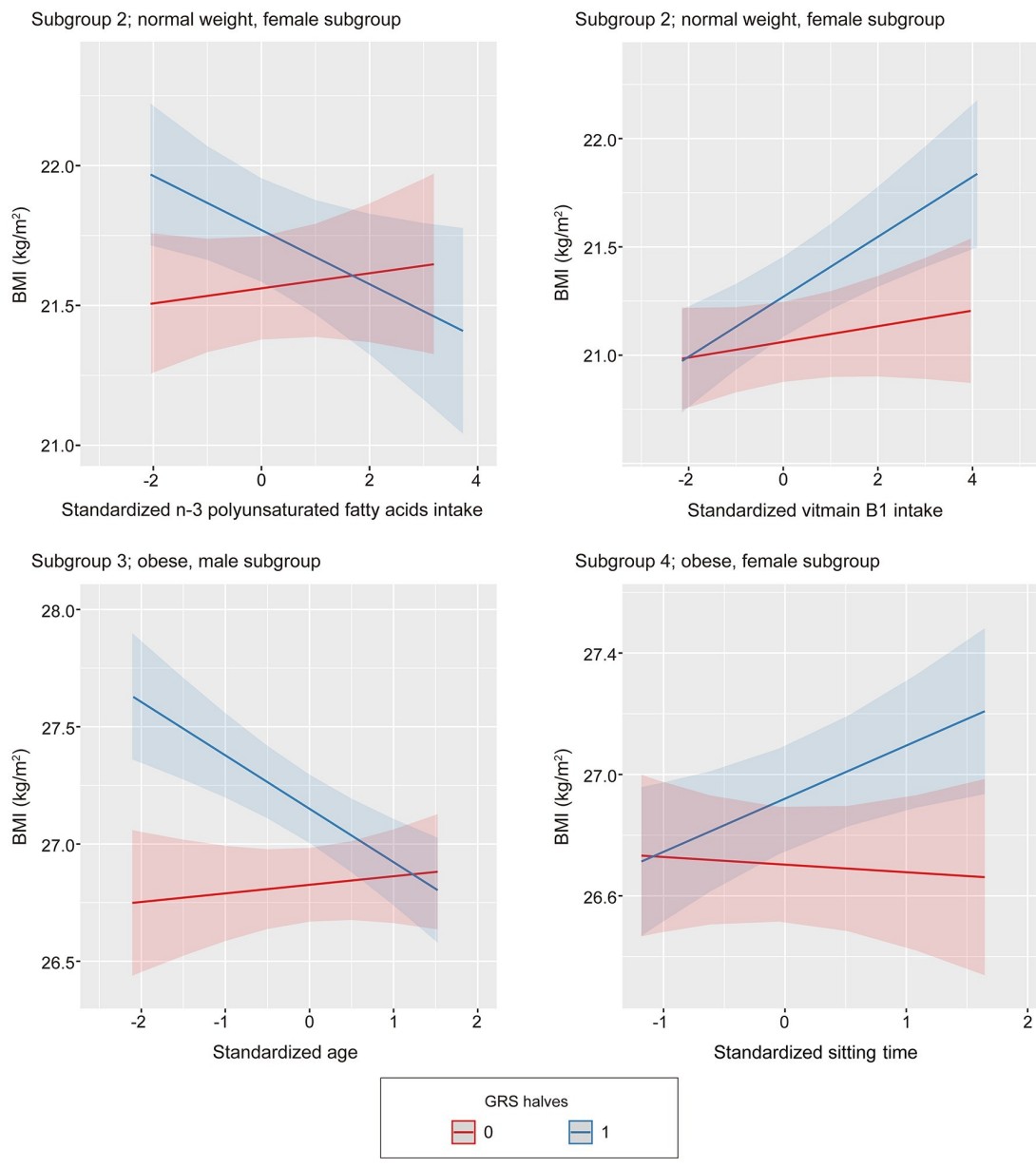

**Fig 2. Subgroup analysis of the interaction between GRS and various lifestyle factors.** Red and blue lines indicate regression lines for each subgroup, i.e., for the lower half (GRS = 0), and upper half (GRS = 1) of GRS. The shading around each regression line shows the 95% confidence interval. One unit of standardized daily intake of *n*-3 polyunsaturated fatty acids, vitamin B1, age, and sitting time corresponds to 0.4 g, 0.07 mg, 8.9 years of age, and 3.5 h, respectively. Mean values for *n*-3 polyunsaturated fatty acids, vitamin B1, age, and sitting time are 2.2 g/day, 0.66 mg/day, 54.7 years, and 4.2 h, respectively. Associations between BMI and each factor differed among groups according to the genetic risk. GRS, genetic risk score.

(sex) and phenotypic information (BMI), which includes basic key tiers considered in personalized therapy [12].

The J-MICC study included multiple lifestyle factors, enabling us to evaluate multiple parameters simultaneously. In particular, we assessed nutritional intake and physical activity in a single model. In addition, both MVPA and sitting time data were available as indicators of physical activity. Although we obtained similar results from the candidate approach, the interaction involving calcium was confounded with saturated fatty acid intake since these are both

highly correlated with milk consumption (S1 Fig). Most previous studies have focused on either nutrition or physical activity, making those studies unable to assess marginal effects that consider confounding effects on each other [4]. Our comprehensive analysis of both nutritional factors and physical activity resulted in the detection of gene–lifestyle interactions, despite somewhat contradictory evidence for such an interaction to date.

A subgroup analysis suggested that stratification according to age, sex, and BMI is necessary for assessing the gene–lifestyle interaction precisely. We constructed four subgroups according to BMI and sex to evaluate interaction effects. Our use of ancestry-specific GRS might explain our ability to detect an interaction in subgroup 4, despite the small sample size, although the interaction effect may have been particularly strong. In subgroup 2, the negative association between $n$-3 polyunsaturated fatty acid intake and GRS was only observed for the high GRS subgroup, similar to the main analysis. Hence, we speculate that a reverse causation exists, i.e., that females with a normal weight that tend to have low weight (low GRS) can intake more saturated fatty acids than those with a high weight (high GRS) because the former group does not have to restrict diet. Consequently, the negative association was observed only in females with normal weight. In this context, differences in the effects of gene–lifestyle interactions depending on factors such as sex and BMI may explain the lack of evidence for interaction effects in the literature.

Our focus on sex and BMI as stratifying factors is based on the previously established difference in BMI between males and females as well as biological characteristics, such as endocrinological (hormonal) characteristics [39]. Furthermore, BMI was selected as a stratifying factor because self-reporting bias is known to be more prevalent in obese subjects [40–43], and the effect of the genetic variants (GRS) is greater for subjects with relatively high BMI [4, 9]. This does not contradict our results indicating that gene–lifestyle interactions are found in both obese subgroups, despite the smaller sample size.

Age was considered as a covariate in the LMM and not a stratifying factor, in part owing to the limited number of participants. Performing the interaction analysis for the subgroups allowed us to identify an interaction between GRS and age in subgroup 3 (males with obesity). Interaction between GRS and age may be explained by the difference in BMI affecting loci and its effect sizes caused by the difference in obesogenic factors with respect to age [44–46]. This raises the hypothesis that genetic variation causes differences in the behavioral response to the obesogenic environment, leading to a difference in BMI [44]. Indeed, the association between GRS and BMI was stronger for relatively younger participants (40–50 years old) within subgroup 3 (Fig 2). Accordingly, studies of GRS or gene–environment interactions should consider the effects of age, sex, and BMI, which may also apply to GWAS of obesity. The female subgroup or the young male subgroup are candidate populations for further analyses of interactions based on our results.

Our study had several limitations. First, as pointed out in a previous methodological review, BMI, an outcome in both the GWAS and our study, does not precisely reflect body composition, such as body fat or lean body mass [5]. Thus, variance in lean body mass, even for the same BMI, may limit our ability to detect associations with obesity. Second, causal inferences were difficult owing to our cross-sectional study design, and the outcome was not the degree of change in BMI. As mentioned above, saturated fatty acids were considered to be observing the reverse causation. Another example was seen for the association between BMI and physical activity among the subgroups; participants in subgroups 1 and 4 were slightly older (S3 Table), and this might have resulted in observing the association between physical activity and BMI only in these subgroups, i.e., individuals who have difficulty in locomotion tend to have a low BMI owing to loss of lean body mass in subgroup 4. In addition, because our study focused on the general population, mainly recruited in

conjunction with specific health checkups [47], participants with a higher BMI had a higher chance of requiring past health guidance (specific health guidance) [47] regarding diet or physical activity. Third, the results are biased due to the self-reporting nature of the study, as nutritional intake, physical activity, and BMI data for about 20% of the participants were self-reported.

## Conclusions

We detected interactions between GRS and nutritional intake and physical activity. Although further study is required to apply these gene–lifestyle interactions in practice, these results provide a basis for the development of optimal prevention or treatment interventions for obesity according to genetic factors, which is expected to substantially improve effectiveness. Further studies of gene–lifestyle interactions stratified by age, sex, and BMI using the degree of change in BMI as an outcome are needed.

## Supporting information

**S1 Fig. Correlation plot between nutrients and foods.** Each number in the matrix indicates correlation coefficients for each pair of nutrients (rows) and foods (columns). *Natto is fermented soybeans, tarako is a salted sack made from pollock or cod roe, chikuwa is a processed fish paste, and ganmodoki is deep-fried tofu fritters. SFA, saturated fatty acids; MUFA, monounsaturated fatty acids; *n*-3 PUFA, *n*-3 polyunsaturated fatty acids; *n*-6 PUFA, *n*-6 polyunsaturated fatty acids; SDF, soluble dietary fiber; IDF, insoluble dietary fiber.
(TIF)

**S2 Fig. *P*-values of the interaction terms observed in the candidate approach.** Each dot indicates the *p*-value for the interaction terms between GRS and each candidate variable in the linear-mixed model. The dependent variable of the model was BMI, with a recruited site-specific random intercept, and the fixed effect was age, sex, GRS, the interaction term between age and sex, and the interaction term between GRS and candidate variable. GRS, genetic risk score.
(TIF)

**S1 Table. BMI-associated loci in the Japanese population.**
(PDF)

**S2 Table. Number of participants recruited at each site.**
(PDF)

**S3 Table. Characteristics of the subgroups of participants according to BMI and sex.**
(PDF)

**S4 Table. Sensitivity analysis excluding participants recruited at Cancer Center.**
(PDF)

**S5 Table. Subgroup analysis by GRS for the candidate approach.**
(PDF)

**S6 Table. Beta coefficients and P-values for the interaction analysis in the candidate approach.**
(PDF)

**S1 Appendix. Supplementary methods.**
(PDF)

## Acknowledgments

We thank Drs Nobuyuki Hamajima and Hideo Tanaka for their work in initiating and organizing the J-MICC Study as former principal investigators. We thank Editage (www.editage.com) for English language editing.

## Author Contributions

**Conceptualization:** Sho Nakamura, Hiroto Narimatsu.

**Data curation:** Sho Nakamura, Xuemin Fang, Azusa Ota, Hiroaki Ikezaki, Chisato Shimanoe, Keitaro Tanaka, Yoko Kubo, Mineko Tsukamoto, Takashi Tamura, Asahi Hishida, Isao Oze, Yuriko N. Koyanagi, Yohko Nakamura, Miho Kusakabe, Toshiro Takezaki, Daisaku Nishimoto, Sadao Suzuki, Takahiro Otani, Nagato Kuriyama, Daisuke Matsui, Kiyonori Kuriki, Aya Kadota, Yasuyuki Nakamura, Kokichi Arisawa, Sakurako Katsuura-Kamano, Masahiro Nakatochi, Yukihide Momozawa, Michiaki Kubo, Kenji Takeuchi, Kenji Wakai.

**Formal analysis:** Sho Nakamura, Xuemin Fang.

**Funding acquisition:** Michiaki Kubo, Kenji Wakai.

**Investigation:** Sho Nakamura, Azusa Ota, Hiroaki Ikezaki, Chisato Shimanoe, Keitaro Tanaka, Yoko Kubo, Mineko Tsukamoto, Takashi Tamura, Asahi Hishida, Isao Oze, Yuriko N. Koyanagi, Yohko Nakamura, Miho Kusakabe, Toshiro Takezaki, Daisaku Nishimoto, Sadao Suzuki, Takahiro Otani, Nagato Kuriyama, Daisuke Matsui, Kiyonori Kuriki, Aya Kadota, Yasuyuki Nakamura, Kokichi Arisawa, Sakurako Katsuura-Kamano, Masahiro Nakatochi, Kenji Takeuchi, Kenji Wakai.

**Methodology:** Sho Nakamura, Xuemin Fang, Yoshinobu Saito, Hiroto Narimatsu.

**Project administration:** Kenji Takeuchi, Kenji Wakai.

**Resources:** Masahiro Nakatochi, Yukihide Momozawa, Michiaki Kubo.

**Supervision:** Hiroto Narimatsu, Kenji Wakai.

**Visualization:** Sho Nakamura.

**Writing – original draft:** Sho Nakamura, Xuemin Fang, Yoshinobu Saito, Hiroto Narimatsu.

**Writing – review & editing:** Sho Nakamura, Xuemin Fang, Yoshinobu Saito, Hiroto Narimatsu, Azusa Ota, Hiroaki Ikezaki, Chisato Shimanoe, Keitaro Tanaka, Yoko Kubo, Mineko Tsukamoto, Takashi Tamura, Asahi Hishida, Isao Oze, Yuriko N. Koyanagi, Yohko Nakamura, Miho Kusakabe, Toshiro Takezaki, Daisaku Nishimoto, Sadao Suzuki, Takahiro Otani, Nagato Kuriyama, Daisuke Matsui, Kiyonori Kuriki, Aya Kadota, Yasuyuki Nakamura, Kokichi Arisawa, Sakurako Katsuura-Kamano, Masahiro Nakatochi, Yukihide Momozawa, Michiaki Kubo, Kenji Takeuchi, Kenji Wakai.

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
