## [Decision Letter · Decision Letter 0]

4 Jul 2022

PONE-D-21-21985Effects of gene–lifestyle interactions on obesity based on a multi-locus risk score: cross-sectional analysisPLOS ONE

Dear Dr. Nakamura,

Thank you for submitting your manuscript to PLOS ONE. After careful consideration, we feel that it has merit but does not fully meet PLOS ONE’s publication criteria as it currently stands. Therefore, we invite you to submit a revised version of the manuscript that addresses the points raised during the review process.

We look forward to receiving your revised manuscript.

Kind regards,

Karen M Davison, PhD

Academic Editor

PLOS ONE

Journal Requirements:

3. Thank you for stating in your Funding Statement: "This study was supported by Grants-in-Aid for Scientific Research for Priority Areas of Cancer (No. 17015018) and Innovative Areas (No. 221S0001) and by the Japan Society for the Promotion of Science (JSPS) KAKENHI Grant (No. 16H06277 [CoBiA]) from the Japanese Ministry of Education, Culture, Sports, Science and Technology. This work was also supported in part by funding for the BioBank Japan Project from the Japan Agency for Medical Research and Development since April 2015, and the Ministry of Education, Culture, Sports, Science and Technology from April 2003 to March 2015."

4. Thank you for stating the following financial disclosure: "This study was supported by Grants-in-Aid for Scientific Research for Priority Areas of Cancer (No. 17015018) and Innovative Areas (No. 221S0001) and by the Japan Society for the Promotion of Science (JSPS) KAKENHI Grant (No. 16H06277 [CoBiA]) from the Japanese Ministry of Education, Culture, Sports, Science and Technology. This work was also supported in part by funding for the BioBank Japan Project from the Japan Agency for Medical Research and Development since April 2015, and the Ministry of Education, Culture, Sports, Science and Technology from April 2003 to March 2015."

Please state what role the funders took in the study.  If the funders had no role, please state: "The funders had no role in study design, data collection and analysis, decision to publish, or preparation of the manuscript.

Reviewers' comments:

Reviewer's Responses to Questions

**Comments to the Author**

1. Is the manuscript technically sound, and do the data support the conclusions?

Reviewer #1: No

Reviewer #2: Partly

2. Has the statistical analysis been performed appropriately and rigorously? 

Reviewer #1: Yes

Reviewer #2: I Don't Know

3. Have the authors made all data underlying the findings in their manuscript fully available?

Reviewer #1: Yes

Reviewer #2: Yes

4. Is the manuscript presented in an intelligible fashion and written in standard English?

Reviewer #1: Yes

Reviewer #2: No

5. Review Comments to the Author

Reviewer #1: This manuscript by Nakamura et al reports on a cross-sectional study assessing the gene environment interaction between a number of dietary and physical activity. This comprehensive reasonably large study includes many useful predictors to assess the impact on BMI.

This reviewer has the following suggestions to improve the manuscript.

Methods:

Study population: It will be helpful if the authors mention that the population was from Japan.

“local population” is obvious for the authors, but not necessarily for a reader outside of Japan. One needs to infer this. They could use: “participants were residents of Japan who participated…..”

How many patients were assessed from the first visit to the cancer hospital. Individuals coming in for cancer visits may or may not be normal. Might these provide confounding for the results. Will it be useful to undertake analysis without these patients to see if there is an impact of these individuals on the results. If there is no impact of these, the authors may include all cases, but it will be useful to report this, as these patients are likely to be different. It is also possible that these patients are already excluded in the outlier removal undertaken in the methods.

Lines 160-164: The physical activity classification using METs is not clear from the explanation provided here. This reviewer tried to read reference # 25, a prior publication of the same cohort, where the levels seem to be different from 5 described here and not straight forward.

Statistical analysis:

Clear and detailed enough to understand the methods. This reviewer is curious to know whether subgroup analysis was performed by sex only. This is frequent in a cohort of this nature, and it will be useful to report even if the results were not informative in such an analysis.

Lines 150-151: Is this the GRS range identified in the cohort? What does the theoretical range mean? If this is indeed the range for the cohort, this should belong to the results section.

Lines 195-196: It is not clear what do the authors mean by candidate approach? Does it mean “univariate” v/s multivariable for the final model? The S4 table suggests use of a multivariable model with selection of predictors. It appears to be subgroup analysis by GRS quartiles. What is the goal of this analysis?

Results:

In general, if the 95% CI is provided in addition to the coefficient, standard error is not required in the table. 95% CI is more informative, and it is suggested that the authors stick with this, rather than SD.

Lines 208-209: This result can be better reported as: “The increase in BMI for every unit increase in the GRS was 2.45 (95% CI xxx-xxx, p <.001).” When one is using weighted GRS, is the unit of increase per risk allele, or per unit increase in GRS?

Table 1: While the table is provided as upper and lower half of GRS, the way to make these subgroups is not clear. Are the 2 subgroups divided at 0? This would be clear if they provided the results of the GRS obtained for the cohort, and clarified that it was divided into 2 groups

It will be useful to provide an overview summary of the results in the tables in 1 or 2 sentences, e.g. there was no difference in the age, sex distribution or the captured environmental variables when compared by the two subgroups of the GRS (Table 1).

Lines 216- onwards for Gene-lifestyle interaction analysis: This section should also include some better language description of the results connecting them with the methods.

e.g. Using the stepwise regression with backward selection, xxx predictors of relevance were selected for the final model.

Line 266: From Table 2, the p-value for the interaction, GRS* saturated fats is 0.016, different from the one in text. Can the authors please verify.

Do the authors have any plausible explanation for the interaction between GRS and age?

In the high GRS group, every year increase in age decreases the BMI by 0.11 (decrease in 1.1/decade)? While this manuscript is focused on the gene-environment effect, this interaction cannot be completely ignored, and needs at least a comment.

Discussion:

This is a cross-sectional study. Hence, the results of the LMM models suggest association. Association is not causation, and it is not appropriate to use the results of the regression models to suggest this.

It is understandable that the results are contrary to the hypothesis for saturated fatty acids. As the authors suggest, it is possible that the negative association may reflect that the recruited patients may be already involved in health counseling. However, it is not appropriate to suggest that this association means that individuals with higher GRS would not benefit from counseling for intake of saturated fatty acids. The analysis from this cross-sectional study will not be able to address such a question. They need to incorporate the results in the context of prior literature.

The authors have selectively addressed the results of the saturated fats interaction with little attention to other results. It is not clear why this was done. There are other lifestyle factors that were statistically significant in the results that need further attention, literature review and better context in discussion.

For wider applicability of results, it may be useful to compare the results of this study with similar studies in different race/ethnic groups.

Reviewer #2: Good job researching on a complex phenomenon of measuring multi-locus risk score for obesity, considering its relevance in the field of personalized genomics. This topic for the manuscript was very well thought out. It is important to look at the combined effect of gene-lifestyle on BMI and its impact on obesity, as it contributes towards realizing the goals of applying an individual’s personalized gene profile to analyze complex phenotypes. The study provides evidence for an interaction effect between genetic risk score on both nutritional intake and physical activi

It is always a good idea to explore through a national multi-institutional cross-sectional data to test for new hypothesis, as this utilizes data that has had effort and time put in to generate it, thereby decreasing the amount of time and work necessary to generate new data. This is also a great step in moving forward to analyze multiple parameters that may have not been tested earlier. Your overall aim is very interesting as it stems from a logical thought process to generate multi-locus risk score considering the polygenic nature of the genetic interactions. However, the hypothesis if there are any or whether it is a hypothesis free research is not clear in the manuscript.

The manuscript is good for consideration for publication, once the suggestions are incorporated.

6. PLOS authors have the option to publish the peer review history of their article (what does this mean?). If published, this will include your full peer review and any attached files.

Reviewer #1: No

Reviewer #2: No

---

## [Author Response · Author response to Decision Letter 0]

31 Jul 2022

Reviewer 1:

Reviewer #1: This manuscript by Nakamura et al reports on a cross-sectional study assessing the gene environment interaction between a number of dietary and physical activity. This comprehensive reasonably large study includes many useful predictors to assess the impact on BMI.

This reviewer has the following suggestions to improve the manuscript.

Methods:

Study population: It will be helpful if the authors mention that the population was from Japan.

“local population” is obvious for the authors, but not necessarily for a reader outside of Japan. One needs to infer this. They could use: “participants were residents of Japan who participated…..”

How many patients were assessed from the first visit to the cancer hospital. Individuals coming in for cancer visits may or may not be normal. Might these provide confounding for the results. Will it be useful to undertake analysis without these patients to see if there is an impact of these individuals on the results. If there is no impact of these, the authors may include all cases, but it will be useful to report this, as these patients are likely to be different. It is also possible that these patients are already excluded in the outlier removal undertaken in the methods.

Lines 160-164: The physical activity classification using METs is not clear from the explanation provided here. This reviewer tried to read reference # 25, a prior publication of the same cohort, where the levels seem to be different from 5 described here and not straight forward.

Statistical analysis:

Clear and detailed enough to understand the methods. This reviewer is curious to know whether subgroup analysis was performed by sex only. This is frequent in a cohort of this nature, and it will be useful to report even if the results were not informative in such an analysis.

Lines 150-151: Is this the GRS range identified in the cohort? What does the theoretical range mean? If this is indeed the range for the cohort, this should belong to the results section.

Lines 195-196: It is not clear what do the authors mean by candidate approach? Does it mean “univariate” v/s multivariable for the final model? The S4 table suggests use of a multivariable model with selection of predictors. It appears to be subgroup analysis by GRS quartiles. What is the goal of this analysis?

Results:

In general, if the 95% CI is provided in addition to the coefficient, standard error is not required in the table. 95% CI is more informative, and it is suggested that the authors stick with this, rather than SD.

Lines 208-209: This result can be better reported as: “The increase in BMI for every unit increase in the GRS was 2.45 (95% CI xxx-xxx, p <.001).” When one is using weighted GRS, is the unit of increase per risk allele, or per unit increase in GRS?

Table 1: While the table is provided as upper and lower half of GRS, the way to make these subgroups is not clear. Are the 2 subgroups divided at 0? This would be clear if they provided the results of the GRS obtained for the cohort, and clarified that it was divided into 2 groups

It will be useful to provide an overview summary of the results in the tables in 1 or 2 sentences, e.g. there was no difference in the age, sex distribution or the captured environmental variables when compared by the two subgroups of the GRS (Table 1).

Lines 216- onwards for Gene-lifestyle interaction analysis: This section should also include some better language description of the results connecting them with the methods.

e.g. Using the stepwise regression with backward selection, xxx predictors of relevance were selected for the final model.

Line 266: From Table 2, the p-value for the interaction, GRS* saturated fats is 0.016, different from the one in text. Can the authors please verify.

Do the authors have any plausible explanation for the interaction between GRS and age?

In the high GRS group, every year increase in age decreases the BMI by 0.11 (decrease in 1.1/decade)? While this manuscript is focused on the gene-environment effect, this interaction cannot be completely ignored, and needs at least a comment.

Discussion:

This is a cross-sectional study. Hence, the results of the LMM models suggest association. Association is not causation, and it is not appropriate to use the results of the regression models to suggest this.

It is understandable that the results are contrary to the hypothesis for saturated fatty acids. As the authors suggest, it is possible that the negative association may reflect that the recruited patients may be already involved in health counseling. However, it is not appropriate to suggest that this association means that individuals with higher GRS would not benefit from counseling for intake of saturated fatty acids. The analysis from this cross-sectional study will not be able to address such a question. They need to incorporate the results in the context of prior literature.

The authors have selectively addressed the results of the saturated fats interaction with little attention to other results. It is not clear why this was done. There are other lifestyle factors that were statistically significant in the results that need further attention, literature review and better context in discussion.

For wider applicability of results, it may be useful to compare the results of this study with similar studies in different race/ethnic groups.

Manuscript: Effects of gene–lifestyle interactions on obesity based on a multi-locus risk score: cross-sectional analysis

General and Scientific comments:

Good job researching on a complex phenomenon of measuring multi-locus risk score for obesity, considering its relevance in the field of personalized genomics. This topic for the manuscript was very well thought out. It is important to look at the combined effect of gene-lifestyle on BMI and its impact on obesity, as it contributes towards realizing the goals of applying an individual’s personalized gene profile to analyze complex phenotypes. The study provides evidence for an interaction effect between genetic risk score on both nutritional intake and physical activi

It is always a good idea to explore through a national multi-institutional cross-sectional data to test for new hypothesis, as this utilizes data that has had effort and time put in to generate it, thereby decreasing the amount of time and work necessary to generate new data. This is also a great step in moving forward to analyze multiple parameters that may have not been tested earlier. Your overall aim is very interesting as it stems from a logical thought process to generate multi-locus risk score considering the polygenic nature of the genetic interactions. However, the hypothesis if there are any or whether it is a hypothesis free research is not clear in the manuscript.

The manuscript is good for consideration for publication, once the suggestions are incorporated.

My overall suggestions would be: 

- To be more descriptive in the introduction and discussion e.g., mentioning about nutrigenomics and its’ application in personalized genomic based preventative therapy. A couple of sentences can be included in the abstract, introduction and discussion. This will add value to the goal of the study and allow for thinking of the "big picture" that could be stem from this kind of research.

- Describe the data a little more clearly rather than simply stating what your figures/tables represent, it will enable to better understand the thought process. This will enable you to interpret your data better and summarize your findings coherently.

- The tables can be presented more precisely and effectively. There is too much information which is not really adding to the overall analysis, e.g., of better presentation is 95% confidence interval can be written in one column rather than splitting it into lower and upper, as this is quite standard and understandable. Representing Mean and SD can be like Mean ± SD rather than Mean (SD), as in the same table in other rows ‘n/ any other parameters’ is written within brackets. The table design can be more simplistic. This will allow your readers to understand the article better. Otherwise on an overall, good analysis and interpretation of statistical data. It can be hard interpreting retrospective data!

- Overall work on the presentation of the figures as well. It will be good to work on the vocabulary of language. There seems to be repeat of words e.g., words like however occur in too many places, synonyms of which can be used. Checking for typo and punctuation errors wherever necessary e.g.,the citations are after the full stop in most of the places. I understand that the authors have used a translation service. It will be better to ensure that the message conveyed is not lost in translation and it is the correct intended one. I will highlight some of these points in the detailed review below.

Specific stepwise comments on the manuscript:

Abstract

- Line 59- To start with importance of lifestyle and obesity research, like introducing about nutrigenomics and then link it to personalized genomics. This will add value.

Introduction

- Same comment as above, to include points about nutrigenomics and then link it to personalized genomics in the first paragraph.

- Line 95- Using ‘contradictory’ rather than ‘controversial’, which is quite a strong word in the context of this area of research.

- Line 101- GRS is not within brackets! It will be better to shift it to line 100 when genetic risk score is first mentioned.

- Line 105- managing obesity is better than the word healthcare. It is too generic a word to be used in this context. 

Materials and methods

- Line 128- DNA was extracted from buffy coat would be more appropriate.

- Line 160- Previously mentioned method, rather than previously method!

- Line 187 to Line 191- This part is unclear. It is mentioned that a sub-group analysis was performed on sex. However, on line 190 and 191, it mentioned as sex was excluded from sub-group analysis. It would be good to verify on this.

Results

- Line 207- Correlation is represented as r, using the symbol for rho would be appropriate.

- Table 1- As mentioned earlier, better to write Mean and SD can be like Mean ± SD rather than Mean (SD). In each row the value within the bracket indicates a different parameter. Could probably find better ways to simplify it! Just a suggestion!

- Describe more about the details of what the table 2 and 3 is conveying in the appropriate place. 

- Table 3- In the ‘age’ row, the values are in fraction. Not sure why the age is being represented in fraction!

Discussion

- Line 299- Change from ‘depending among populations’ to “depending on populations”

- Line 302 & 303- Not clear what is being indicated here, is it the suggestion for the entire findings or just the previous sub-group 4 findings. Also, changing ‘GWAS of the Japanese population’ to “GWAS on the Japanese population” is suggested.

- Line 310 & 312- Not clear interpretation, adding more details will add value. Mentioning contradictory is as controversial gives a very different tone.

- Line 329-Better to replace ‘a relative’ with “relatively”

- Line 339- This sentence “younger participants (40–50 years old) in our analysis of subgroup 3” is misleading. Age 40-50 is considered middle age. If it is in comparison, then better to state relatively younger to the other sub-group!

- Line 347-349- The sentence ‘Other than saturated fatty acids, the age of participants differed slightly among subgroups …..’ can be framed better. 

- Line 347-349- The sentence ‘mainly recruited at health checkups,…….’ is not clear. Not sure why past health guidance is mentioned.

---

## [Decision Letter · Decision Letter 1]

2 Dec 2022

Effects of gene–lifestyle interactions on obesity based on a multi-locus risk score: a cross-sectional analysis

PONE-D-21-21985R1

Dear Dr. Nakamura,

We’re pleased to inform you that your manuscript has been judged scientifically suitable for publication and will be formally accepted for publication once it meets all outstanding technical requirements.

Kind regards,

Tomoyoshi Komiyama, Ph.D

Academic Editor

PLOS ONE

Additional Editor Comments (optional):

Dear Authors,

Thank you for submitting your revised manuscript.

Your study analyzed how genetic risk and lifestyle factors (such as nutritional intake and physical activity) are linked to BMI, based on cross-sectional data from the Japan Multi- Institutional Collaborative Cohort Study.

I see you corrected the highlighted sections according to the reviewer’s suggestions.

Also, I found your updated version much clearer than the original.

Therefore, I have no further comments as all of my previous concerns were adequately addressed.

I believe this manuscript will satiate the reader's interest.

Tomoyoshi Komiyama

Reviewers' comments:

Reviewer's Responses to Questions

**Comments to the Author**

1. If the authors have adequately addressed your comments raised in a previous round of review and you feel that this manuscript is now acceptable for publication, you may indicate that here to bypass the “Comments to the Author” section, enter your conflict of interest statement in the “Confidential to Editor” section, and submit your "Accept" recommendation.

Reviewer #2: All comments have been addressed

2. Is the manuscript technically sound, and do the data support the conclusions?

Reviewer #2: Yes

3. Has the statistical analysis been performed appropriately and rigorously? 

Reviewer #2: I Don't Know

4. Have the authors made all data underlying the findings in their manuscript fully available?

Reviewer #2: Yes

5. Is the manuscript presented in an intelligible fashion and written in standard English?

Reviewer #2: Yes

6. Review Comments to the Author

Reviewer #2: Appreciate the authors for incorporating all of the suggested changes. It would be good to review the article for copy-edits and formatting as required by the Journal.

7. PLOS authors have the option to publish the peer review history of their article (what does this mean?). If published, this will include your full peer review and any attached files.

Reviewer #2: No

---

## [Editor Report · Acceptance letter]

30 Jan 2023

PONE-D-21-21985R1 

Effects of gene–lifestyle interactions on obesity based on a multi-locus risk score: a cross-sectional analysis 

Dear Dr. Nakamura:

I'm pleased to inform you that your manuscript has been deemed suitable for publication in PLOS ONE. Congratulations! Your manuscript is now with our production department. 

Kind regards, 

on behalf of

Dr. Tomoyoshi Komiyama 

Academic Editor

PLOS ONE